# Teaching Communication and Functional Life Skills in Children Diagnosed with Autism Spectrum Disorder

**DOI:** 10.3390/bs15020198

**Published:** 2025-02-12

**Authors:** Juliana Ribeiro Rabelo Costa, Daniel Carvalho de Matos

**Affiliations:** 1Postgraduate Program in Psychology, Campus do Bacanga, Federal University of Maranhão, São Luís 65080-805, MA, Brazil; rabelo.juliana@discente.ufma.br; 2Center for Teaching, Research in Psychology, Inclusive Education and Health, Evoluir Institute, São Luís 65075-690, MA, Brazil; 3Department of the Undergraduate Course in Psychology, Campus Renascença, CEUMA University, São Luís 65075-120, MA, Brazil

**Keywords:** autism spectrum disorder, script fading, applied behavior analysis, repertoires, say-do correspondence

## Abstract

Children with autism spectrum disorder (ASD) commonly show difficulty in communication and daily functional skills. The use of scripts may help establish these repertoires. Scripts may be visual (e.g., pictures depicting actions), textual (e.g., printed or written sentences depicting actions), or auditory (e.g., recorded or dictated phrases depicting actions). Background/Objectives: The purpose was to assess the efficacy of script fading in establishing the vocal verbal emission of sentences under the control of pictures representing actions from four behavioral sequences (e.g., brushing teeth) in three children with ASD. The effects of the intervention on the emergence of related non-verbal repertoires were evaluated. During intervention, the scripts were textual for one participant, who initially read the sentences. For the remaining two participants, scripts were dictated to them so they could repeat them. Across sessions, scripts were faded out by gradually omitting the words from the sentences. Results: Script fading produced the emission of sentences solely in the presence of pictures (tacts according to a Skinnerian approach of language), replicating a previous study in which the same procedure also established the same type of repertoire. However, as an extension, in the current investigation, related non-verbal actions also emerged. Other previous studies into script fading were not specifically concerned with teaching tacts and probing the emergence of related non-verbal untaught repertoires. Conclusions: The data were interpreted as indicating correspondence between verbal and non-verbal behavior or “say-do” correspondence. The data were discussed in the sense that script fading, for some learners, may improve communication with sentences that impact the acquisition of related non-verbal behavioral chains. Limitations of the research were discussed.

## 1. Introduction

In applied behavior analysis (ABA), methodologies and technologies of teaching are developed to solve human issues, including the treatment of children diagnosed with autism spectrum disorder (ASD). Regarding this public in general, ABA interventions are frequently addressed to establish socially significant repertoires, which are commonly impaired in many learners ([1]; [2]; [3]; [14]; [20]). Careful assessments of skills related to communication, conversation, and other important verbal and non-verbal repertoires should be conducted, helping to support intervention practices to ameliorate possible deficits (see, for example, [6]). Regarding interventions to address deficits in individuals with atypical development, among the variables which bolster development, visual scripts may be used to establish behavioral sequences related to daily functional activities such as getting dressed, brushing teeth, preparing a meal, etc. Visual scripts may consist of pictures or phrases depicting functional actions while performing a task (e.g., grabbing a toothbrush and toothpaste, adding toothpaste on the toothbrush, etc.). In a study conducted by [19] ([19]), visual scripts (pictures depicting sequences of actions) were used to teach three children with ASD between 6 and 9 years old to perform functional daily skills independently. Three tasks were defined for each learner (e.g., setting the table, making lunch, and washing clothes). After a baseline condition showed that the children were unable to perform the tasks, an intervention condition was implemented in three phases. First, everyone learned to receptively discriminate between the pictures representing the actions of each defined behavioral sequence.

After this, the learners were taught to choose a preferred reinforcer, turn pages of a book with pictures, perform motor actions portrayed by the pictures, and reinforce themselves. In the end, the presence of an interventionist faded out so that the tasks could be performed during his absence. As a result, all children were able to finish the defined tasks. In addition, there was evidence of a reduction in the frequencies of disruptive behaviors (stereotypes). For one child, training took place in a clinic room. For the remaining two children, training occurred in a home setting. After intervention was successful for each learner in a specific environment, generalization measures, across the defined tasks, were obtained through probes in a new environment (either clinic or home depending on the child). Post-training probes also showed that the pictures were exerting control over the behaviors of two of the children (there were changes in the order of the pictures in the book during these probes) ([19]). Later, other studies involving visual scripts were also conducted to establish sequences of actions concerning functional play. [17] ([17]) used visual scripts in three 4-year-old children with atypical development (one with ASD and two with language delays).

Four sets of toys were organized (e.g., a barn with a horse, cow, tractor, etc.). Five actions were defined for each set and were represented by pictures in a binder/book. After the participants learned the prerequisites (e.g., receptively identifying pictures and turning binder pages), a baseline showed that the children could not perform the sequences of actions related to each toy set. An intervention was then implemented, and it consisted of a least-to-most hierarchy of prompts provided by an experimenter. This consisted of the following steps: (1) general instruction to perform the action depicting a given picture; (2) specific instruction to perform an action; (3) pointing to relevant objects during the provision of a specific instruction; (4) modeling a response during a specific instruction; and (5) provision of physical guidance. Independent responses produced praise and sometimes access to preferred items. The intervention was effective for all participants. There was evidence of generalization across new sets of toys not directly trained, and it was suggested that the pictures exerted control over the actions (through probes with changes in the order of the pictures) ([17]).

[18] ([18]) conducted two systematic replication experiments (involving three children with ASD between 4 and 9 years old in the first experiment and three other young learners between 7 and 19 years old, two diagnosed with ASD and one with Down syndrome, in the second experiment) of the previous investigation by [17] ([17]). However, textual instructions (phrases for oral reading) were used instead of pictures. In each experiment, four sets of toys and five different actions for each set were programmed. Each action to be emitted was represented by a textual instruction (printed phrase) that the learners had to follow. During a baseline condition, a board with numbered printed instructions were shown for carrying out the actions for each set of toys. During intervention, correct responses (performing the actions represented by the instructions) were reinforced. When necessary, an experimenter used a least-to-most prompting hierarchy similar to Phillips and Vollmer.

After a learning criterion was achieved for a given set of toys, probes with the textual instructions under a new order/sequence were conducted. If a learner was not able to perform the actions, a new teaching condition was defined and, in this case, the textual instructions would be presented one at a time on the board (instead of all at the same time). As a learning criterion was achieved, new probes to assess responses with the textual instructions under a new order/sequence would be conducted. During the second experiment, more specifically, orienting responses to the instruction board were evoked (e.g., looking at the board, following the texts with the hand or reading the texts out loud). A test condition was defined after baseline to determine if the learners would be able to perform actions solely under control of a general vocal instruction (e.g., “do what is being said here”). In addition, for one learner, there was a teaching condition during which the textual instructions were presented twice or three times at once ([18]).

As a result, regarding both experiments ([18]), all learners needed to reach some training level to be able to emit all actions from each set of toys. Everyone also was able to perform the non-trained second sequence of actions after training the first sequence for each of the toy sets. The authors discussed the data as possible indicators of generalization regarding following instructions for the same set (new order/sequence of instructions), and that the textual instructions were in fact exerting control over the actions performed by the learners. Nevertheless, nobody showed a generalized repertoire of following textual instructions across different sets of toys. It is possible that generalization data would be more robust if the participants had better reading skills.

Using visual scripts and other variables has also been important to teach behaviors to prevent contagion from infectious diseases such as COVID-19. This may be particularly relevant for children diagnosed with ASD who commonly have difficulty performing behavioral sequences. [8] ([8]), for example, conducted a review study on behavioral strategies to teach the repertoire of washing hands. Some strategies were based on antecedent events such as, for example, instructions for modeling washing hands properly or providing vocal and visual prompts/scripts. Other strategies addressed the manipulation of consequence events such as visual feedback, error correction, and differential reinforcement. Another study concerned with the prevention of diseases was conducted by [13] ([13]). They remotely (through the internet) trained parents to use visual (pictures) and textual (phrases representing actions) scripts to teach four children with ASD between 4 and 10 years old to put on surgical masks and sustain their use for an extended period (a behavioral chain concerning ten steps to be performed). Physical guidance was provided when needed until the learners could perform the behavioral sequence in the presence of the visual scripts. Thereafter, the scripts gradually faded out. The intervention was successful and, later, the children were taught to keep masks on their faces for a longer period. This was achieved using verbal and gestural prompts and differential reinforcement of other behaviors except removing the masks.

Research involving scripts has also been concerned with the development of communication or, in other words, verbal behavior ([21]). The current investigation was particularly consistent with the literature that focused on the use of visual scripts consisting of pictures and/or sentences. A seminal study was conducted by [9] ([9]). These authors assessed four children with ASD, who were fluent readers, in the context of activities with groups of learners. The children remained in the same room, each accompanied by an interventionist, and had opportunities to play and verbally interact with each other. All children also understood actions related to games and fun. To help the learners in the process of communicating with each other, textual scripts were organized with statements and questions about actions in the present, past, and future tenses (e.g., “did you enjoy playing on the swing yesterday?”). Baseline showed that the children were unable to vocally interact using sentences independently.

As a baseline, they rarely initiated communication with each other, although they all had some functional expressive language and, sometimes, sought out important adults spontaneously. Still during baseline, painting, drawing, and coloring activities were also performed by the children. It was at this moment that each one of them was told to play and talk. During the intervention condition, at first, the children had to read the scripts in the process of interacting with each other. When necessary, they were physically guided by an interventionist to pay attention to the scripts. After reading the scripts accurately, five fading steps were established by gradually removing/omitting the words from sentences from the end to the beginning. The purpose of this measure was that, at some point, the learners would be able to talk free from any script control and this was demonstrated for all children. There was also evidence of response generalization in the presence of a new communication partner, environment, and materials. Performance maintenance was also demonstrated two weeks later ([9]).

In a later study, [10] ([10]) used script fading involving pictures and short phrases to establish communication in three children diagnosed with ASD with more limited reading skills at school. Before the research, all learners only spoke in the context of answering questions or demanding access to edibles and preferred toys. During the baseline, the children did not verbally communicate with a teacher in a functional manner. During intervention, textual scripts (“look” and “watch me”) to evoke the attention of a communication partner (a known teacher) were incorporated into the children’s photography activity schedules with toys. All children were fluent in performing actions depicted by pictures. After the textual scripts were faded out, the unscripted interactions with the teacher continued and generalized to different activities.

[11] ([11]) systematically replicated the research by [9] ([9]). Script fading was used in three children with ASD with the purpose of improving communication and interaction among them and with adults during a social activity (e.g., group games with toys). Sessions occurred in a university clinic. During the baseline, the children obtained up to four sets of toys. Each set comprised three similar toys. When each child was provided with a given set, she was told to share it with the others. After 10 s, the children were instructed to return the toys from the set and the interventionist provided a new set, repeating the procedure. During the baseline, the learners hardly ever talked among them. During intervention, besides the access to toy sets for sharing as in the baseline, an interventionist provided textual scripts for each child to communicate. The scripts included phrases that would be suitable for starting a game or communicating with a peer while sharing toys. When needed, the interventionist provided help to read the scripts. After the reading performance was fluent, the words from the scripts gradually faded out. As a result, all children communicated better and without the need for reading scripts.

More recently, a study on script fading involved four children with ASD between 5 and 10 years old ([15]). They were unable to describe non-verbal stimuli such as pictures through vocal verbal sentences comprising four or more words (e.g., saying “the car has four wheels” under the picture of car and instruction “tell me something about the car”) as it was shown in a baseline condition. According to a Skinnerian ([21]) approach to language, the type of verbal repertoire contemplated in the research is called tact and comprises the emission of a vocal verbal response under the control of a non-verbal discriminative antecedent stimulus. Responding is maintained by an established form of attention (e.g., verbal praise), that is, a generalized conditioned reinforcer. Two of the children exhibited a wide repertoire of sentence reading, and the other two children did not. These, however, demonstrated echoic of sentences at a generalized level. During the intervention condition, for the learners who read fluently, pictures and written sentences were used so they could read. For the remaining learners, the same pictures and written sentences were used as well, but an experimenter read them out loud, so the children could repeat/echo them.

When the children consistently read the sentences (or echoed them under the experimenter’s vocal model), script fading was defined in five steps. Along the steps, the words were gradually removed/omitted so, with time, all children were able to vocally emit the sentences solely under the control of pictures and the verbal instructions to describe them. This was systematically demonstrated by each learner across three different groups of five pictures. In addition, a follow up probe session was conducted one month later (after the intervention was discontinued) and all children demonstrated errorless performance. A limitation pointed out by the authors was that varied scripts were not used, that is, the sentence assigned to each picture was always the same. It was discussed that future replication studies should use varied sentences and assess how this impact learners’ performance ([15]).

The literature on the use of scripts (and their fading) to teach skills to children diagnosed with ASD shows that this procedure is effective in the acquisition of important non-verbal and verbal repertoires. However, new investigations should be carried out to assess if teaching one type of repertoire using script fading may result in the emergence of another related repertoire. The current study aimed to evaluate if script fading would effectively establish a repertoire of emitting sentences with tact function in children with ASD (systematic replication of the research by [15]). Also, the effects of the intervention were measured on the emission of non-verbal repertoires to which non-verbal stimuli (scenes) and corresponding sentences were related.

## 2. Materials and Methods

### 2.1. Participants

This study involved three children diagnosed with ASD, aged between 4 and 6 years old, as participants (P1, P2, and P3). They all had a history of receiving ABA-based interventions in a clinical or residential environment. As inclusion criterion, the participants should not be able to emit descriptions of the research materials (scenes depicting behavioral sequences) consisting of sentences with four or more words. However, they already demonstrated a reading or vocal imitation (echoic) repertoire of sentences at a generalized level, as well as generalized instruction following. They were also able to select many non-verbal stimuli as listeners (e.g., by touching pictures in arrays under the control of their spoken names), and to pair many visual stimuli (by relating equal and similar objects and pictures among themselves). Children who did not follow all the requirements described so far would be excluded from the study. It is important to mention that, among the selected participants, only P2 showed a repertoire of reading words and sentences at a generalized level. P2 was also able to pair printed words and sentences to corresponding pictures/scenes.

### 2.2. Environment

Data collection was conducted in a structured room, with a table and three chairs, from a private clinic (under the consent of the director of the institution) and the participants’ own residences under the due authorization of the parents or other guardians. In other words, the environments were those where the children used to receive specialized interventions on ABA regardless of the research. However, data collection only took place at different times and as authorized by those responsible for the children.

### 2.3. Interobserver Agreement and Fidelity of Procedures

During data collection, an experimenter and a given participant sat at the table facing each other. On several occasions, the experimenter administered assessment and teaching trials of repertoires to the participant. Data were recorded on personalized record sheets with pencil and entered into an Excel spreadsheet for tabulation and analysis of the degree of agreement between the experimenter and a second observer who, occasionally, also took data. This second person, blind to the research objectives, collected data in 30% of the study regarding the different stages of baseline, probes, teaching, and follow up through videos recorded by the experimenter. Along several sessions, the interobserver agreement (IOA) was calculated by dividing the number of trials with agreement by the total number of trials. Then, the result was multiplied by 100 to obtain a percentage. The average agreement, considering the three participants, was 99% for P1, 98% for P2, and 100% for P3.

To obtain fidelity measures on the implementation of research procedures, a checklist was used by an independent observer, and it described behavioral components whose correct implementation by the experimenter represented high fidelity of the procedures (e.g., correcting stimuli manipulation during assessment and intervention sessions, waiting at least 5 s for the participant to emit a response in a trial, reinforcing correct participants’ responses during intervention trials, and implementing correction procedures when the participants committed errors in intervention trials). In each session, the number of components implemented correctly by the experimenter was divided by the total number of components. The result was multiplied by 100 to obtain a percentage, which was 100% during assessment and intervention sessions with each participant.

### 2.4. Materials

The experimenter used a personalized sheet and pencil to systematically record responses emitted by the participants in assessment and teaching trials. Non-verbal stimuli were used consisting of pictures/scenes depicting behavioral sequences of (1) putting on a mask and keeping it on the face, (2) cleaning up the hands with hand sanitizer, (3) playing functionally with a Mr. Potato Head, and (4) performing daily living skill of brushing teeth. For each action from the behavioral sequences, besides the picture, a printed sentence to be read or echoed by the participant (one case or the other depending on his/her repertoire) was also used. Table 1 shows all sequences, materials/items, and scripts/sentences used in the research stages. The sequence pictures can be found in Appendix A.

### 2.5. Dependent Variables and Independent Variables

The main dependent variable (DV) consisted of the emission of sentences with tact function ([15]) related to pictures from the four programmed behavioral sequences for each participating child. Another DV corresponded to the emission of related non-verbal responses portrayed by the pictures (and verbalizations) from the sequences. The independent variables (IVs) corresponded to differential reinforcement of independent responses, consisting of sentences with tact function and script fading (regarding scripts that were read by the child or modeled by an experimenter so the child could echo). One child (P2) participated in a teaching condition involving scripts for reading because he could read sentences of varying lengths fluently. The remaining children (P1 and P3) participated in a teaching condition involving scripts that were dictated to them, so they could echo.

### 2.6. Experimental Design

In the current study, the emission of sentences with tact function (main dependent variable—DV) was directly taught under the control of scenes/pictures depicting the following behavioral sequences: (1) cleaning up the hands, (2) putting on the mask and keeping it on the face, (3) functional play (Mr. Potato Head doll), and (4) brushing teeth. Also, the effects of the intervention were measured on the emission of non-verbal repertoires to which the scenes and corresponding sentences were related (secondary DV). If these repertoires did not emerge as a function of teaching the emission of tacts, they would also be directly taught through a least-to-most prompting hierarchy and differential reinforcement of independent performance, as it was the case of previous studies ([17]; [18]).

To ensure the establishment of experimental control of the independent variable (IV) (teaching the emission of sentences with tact function through script fading and differential reinforcement of independent performance) on the main DV mentioned, a single case research design consisting of a concurrent multiple baseline design ([3]) across different behavioral sequences for each participant was used. Initially, each participant was assigned to a probe session to assess the vocal verbal emission of sentences under the control of pictures from the behavioral sequences (e.g., pictures representing the step-by-step sequence of brushing teeth) and, also, a probe session to assess the emission of non-verbal repertoire of emitting the actions related to the pictures (e.g., putting toothpaste on the toothbrush in presence of the picture representing this action). Once the participants did not demonstrate these skills during the initial probes, a baseline condition (concerning more sessions to assess the emission of sentences with tact function) was administered for each of the four behavioral sequences. After the stability of data representing low performance was achieved, a gradual script fading intervention condition to teach the emission of tacts was established for one of the four behavioral sequences (while baseline sessions continued for the remaining sequences).

Once the repertoire of emitting sentences with tact function was established for the first behavioral sequence in the absence of scripts (either textual or dictated scripts for reading or echoic responding, respectively, depending on the participant), maintenance training was applied, and the intervention was also administered for the next sequence (second). In parallel, the baseline condition continued for the third and fourth behavioral sequences. After a child acquired the tact repertoire for the second behavioral sequence, maintenance training was applied and the intervention also commenced for the third sequence. The fourth and last sequence continued in baseline until a learning criterion of the tact repertoire was also achieved for the third sequence (then, maintenance training began for the third sequence, and the intervention was also administered for the fourth sequence until acquisition of the target skill). It is worth noting that whenever the learning criterion of tact training with a given behavioral sequence was reached, a new probe session to assess the possible emergence of non-verbal repertoires related to pictures and verbalizations from the sequences was also conducted.

### 2.7. Procedure

The research stages are presented in Table 2 and described thoroughly next.

First stage (P2 from the condition with textual scripts): Initial probe and baseline of emission of sentences with tact function (first repertoire) and initial probe for the emission of non-verbal related responses (second repertoire). The experimenter presented four behavioral sequences for P2. Each sequence consisted of non-verbal stimuli (pictures) representing several actions. In the presence of each picture, when the experimenter said, “tell me what is happening here”, P2 had up to 5 s to make a description (e.g., saying “put toothpaste on the toothbrush” under the control of its corresponding picture). The pictures were organized in a binder (one picture per page) and P2 could already turn the pages independently. In the initial probe session with each sequence, the number of trials to respond depended on the number of pictures representing the sequence. In parallel, one initial probe session for the emission of non-verbal responses concerning the pictures from the routines (e.g., the child put toothpaste on the toothbrush after seeing the related visual stimulus) was conducted. At this moment, the experimenter provided the binder to P2, as well as all necessary items to perform the target responses. When told to perform a given behavioral sequence (e.g., “show me how you brush your teeth”), P2 had up to 30 s to emit the actions from the sequence using the binder with pictures. During this period, the experimenter oriented P2 to look at each picture in the binder. After the initial probe session for the two mentioned repertoires, a baseline condition concerning the first (sentences with tact function) was administered for each behavioral sequence. The only difference compared to the initial probe was that baseline involved several sessions (until stability with low performance was demonstrated). It is worth noting that, in all probe and baseline sessions concerning the two repertoires at this stage, differential consequences for correct and incorrect responses were not provided.

Second stage (P2 from the condition with textual scripts): Teaching script reading. For each behavioral sequence, the number of trials to be delivered per training session depended on the number of pictures from the sequence. Each picture from a behavioral sequence represented a trial to respond to. It is important to mention that, at this stage, each picture was accompanied by a related printed sentence and that P2 had to read out loud. In each trial from a session, the experimenter presented the verbal instruction “read” and P2 had up to 5 s to read. When a correct response was emitted, verbal praise and a token were delivered (cumulative tokens were, by the end of each intervention session, exchanged for a preferred item such as a toy or video appropriate for the participant’s age. This was also applied to sessions during the next intervention stage). When an error occurred, or no response was emitted in 5 s, a correction procedure was administered, that is, the experimenter read the sentence so P2 could repeat the information. The termination criterion for this stage of the study, for each behavioral sequence, consisted of two sessions without errors.

Third stage (P2 from the condition with textual scripts): Script fading (fading of textual scripts). In this stage, unlike the previous one, textual scripts gradually faded out, that is, the words from the printed sentences concerning each behavioral sequence were gradually removed (from the end to the beginning of each sentence). The number of fading steps varied between three and four steps across the sentences from the behavioral sequences. The sentences were organized in a manner that the number of words was balanced (regarding the original words in Brazilian Portuguese). As an example of fading steps, during first step of the sequence to clean up hands with hand sanitizer, in each sentence, the last word was omitted (e.g., “Apply the hand sanitizer to your …”). In this case, P2 had to textually respond (read) to the visible information and “fill in the blank”, that is, verbalize the missing word (e.g., “hand”). When P2 verbalized what was missing, he was praised and received a token. When an error occurred, or no response was emitted in 5 s, the experimenter showed the previous sentence including the missing word, so the child could read. The transition to each of the following fading steps occurred as soon as the participant did not commit errors in two consecutive sessions. Along the fading process, once P2 managed to verbalize all sentences solely under the control of the pictures, this stage was terminated.

Fourth stage (P2 from the condition with textual scripts): Additional probes to assess the emergence of the emission of non-verbal responses concerning the behavioral sequences. Throughout the research, additional probe sessions were administered to assess if non-verbal responses would emerge as a function of systematically training the emission of sentences with tact function. The additional probe sessions were conducted in a manner which was identical to what happened during the initial probe from the first stage of the research. Differential consequences for correct and incorrect responses were not provided as well.

Fifth stage (P2 from the condition with textual scripts): Follow up (two weeks after the end of research). Two weeks after the end of the fourth stage, a probe was conducted to assess the emission of sentences with tact function. The task was identical to the initial probe from the first stage. The research stages for the remaining participants (P1 and P3), to whom scripts were dictated for the emission of echoic responses during the teaching of tacts, are described in detail next.

First stage (P1 and P3 from the condition with dictated scripts for the emission of echoics): Initial probe and baseline of emission of sentences with tact function and initial probe for the emission of non-verbal related responses. This stage was identical to the one conducted for P2.

Second stage (P1 and P3 from the condition with dictated scripts for the emission of echoics): Teaching the emission of echoics under the control of dictated scripts. This case was also similar to the second stage for P2. In each trial from a session, the picture also involved a related printed sentence, but this time, the experimenter was the one who read the sentence so a given participant could repeat (echo) it. Up to 5 s were allowed to respond. An independent response of echoing the whole dictated sentence was followed by verbal praise and a token. When an error occurred, or no response after 5 s, the experimenter dictated the sentence again as a second opportunity for the learner (however, only a correct response in the first opportunity was considered to reach the learning criterion). The learning criterion for each behavioral sequence consisted of two sessions without any errors.

Third stage (P1 and P3 from the condition with dictated scripts for the emission of echoics): Script fading (fading of dictated scripts). The characteristics of this stage were similar to those from the third stage for P2 with textual scripts. In the current cases of P1 and P3, the dictated scripts gradually faded out, that is, the words from the sentences were gradually omitted (from the end to the beginning of each sentence). During the first fading step, the last word from each sentence representing a given behavioral sequence was omitted, that is, the experimenter dictated the sentence until the moment of omission of information. The participant, then, had up to 5 s to “fill in the blank”, that is, to verbalize what was omitted. An independent response resulted in verbal praise and a token. When an error occurred, or no response was emitted within 5 s, the experimenter vocally provided the information omitted so the learner could repeat it. The transition to each of the following fading steps occurred as soon as the participant did not commit errors in two consecutive sessions. During the fading process, once P1 and P3 managed to verbalize all sentences solely under the control of the pictures, this stage was terminated.

Fourth stage (P1 and P3 from the condition with dictated scripts for the emission of echoics): Additional probes to assess the emergence of the emission of non-verbal responses concerning the behavioral sequences. This stage was identical to the fourth stage for P2 with textual scripts.

Fifth stage (P1 and P3 from the condition with dictated scripts for the emission of echoics): Follow up (two weeks after the end of research). This stage was identical to the fifth stage for P2 with textual scripts.

### 2.8. Ethical Procedures

This study is related to a project, which was approved by an ethics committee in research with humans (authorization No. 6.585.992) from Federal University of Maranhão, Brazil, São Luís-MA. The participants’ parents signed an informed consent form. The participants themselves also signed an informed consent form. The identities were kept confidential, and participation could be terminated at any time and without any harm.

## 3. Results

For each of the three participants (P1, P2, and P3), the data were organized in the following manner: Data concerning the performance in the repertoires of emitting sentences with tact function and related non-verbal actions under the control of pictures from the behavioral sequences. Data on non-verbal actions were only collected in probes before and after teaching the emission of sentences with tact function.

The graphs from Figure 1, Figure 2 and Figure 3 were organized to represent performances of P1, P2, and P3, respectively, in the mentioned target repertoires. During intervention to teach the emission of sentences, the script fading was established for each of the four behavioral sequences based on the number of words determined for each sentence. For example, three fading steps (F1–F3) were defined for each sentence of the first sequence (cleaning up the hands). The unfaded first sentence, for example, corresponded to “take the bottle with hand sanitizer”. The fading steps of this case were organized in the following manner: F1—“take the bottle with …”; F2—“take the…”; and F3 (all words from the sentence omitted). Figure 1, below, presents the results of participant P1.

One single initial probe session for tacts was administered for each behavioral sequence. In parallel, one initial probe session was administered for the related non-verbal actions (e.g., the child had to take the bottle with hand sanitizer after seeing the picture representing this non-verbal action). According to data shown in Figure 1, P1 did not show any correct responses in the initial probes for both repertoires in all sequences. The same case applied to the baseline (performance was stable across several sessions). During intervention with the first behavioral sequence (cleaning up the hands), five sessions involving five trials each were necessary for P1 to reach the learning criterion during the condition with unfaded scripts. When script fading commenced, P1 needed three, four, two, and four sessions to achieve criterion in F1, F2, F3, and F4 fading steps, respectively. In the end, P1 was able to emit all sentences solely under the control of pictures from the first behavioral sequence.

After P1 was able to describe all pictures from the first sequence without scripts, two additional probe sessions to assess non-verbal actions (AP-NVAs) were administered for each of the four behavioral sequences. P1 was not able to emit the non-verbal actions under the control of the pictures. Then, maintenance tact training of the first sequence was applied. After that, all non-verbal actions emerged for the first sequence. These data were replicated across additional sessions (maintenance tact training–probing related non-verbal actions). Independent performance concerning both tact and related non-verbal repertoires was also demonstrated in follow up sessions for the first sequence. In parallel with this, tact training with unfaded scripts (USs) was applied for the second behavioral sequence (putting on the mask and keeping it on the face).

Three sessions were needed during the US condition. Across script fading steps, P1 needed three, four, nine, and three sessions to achieve criterion in F1, F2, F3, and F4, respectively, for the second sequence. Then, two AP-NVA probe sessions were administered for the second, third, and fourth behavioral sequences. As it may be seen in Figure 1, P1 was able to correctly emit all non-verbal actions regarding the second sequence. In addition, these data were replicated across several additional sessions (maintenance tact training–probing related non-verbal actions). Plus, independent performance concerning both tact and non-verbal repertoires was shown in follow up sessions.

Parallel to the acquisition of the repertoires from the second sequence, tact training concerning the third behavioral sequence (functional play) commenced. Three, two, two, and two sessions were needed for P1 to achieve a learning criterion in US, F1, F2, and F3, respectively. Two AP-NVA probe sessions were administered, and P2 did not show any errors. In addition, independent performance concerning the repertoires was replicated across additional sessions (maintenance tact training–probing related non-verbal actions), as well as it was demonstrated across follow up probe sessions.

Parallel to the acquisition of the repertoires from the third sequence, tact training commenced with the fourth sequence (brushing teeth). P1 needed three, two, four, and five sessions to reach criterion in US, F1, F2, and F3, respectively. Two AP-NVA probe sessions were conducted, and P1 showed no errors. In addition, as it was the case of the previous sequences, independent performance in both repertoires (maintenance tact training–probing related non-verbal actions) was verified, as well as it was demonstrated across follow up probe sessions. Next, Figure 2 shows data from P2.

**Figure 2 behavsci-15-00198-f002:**
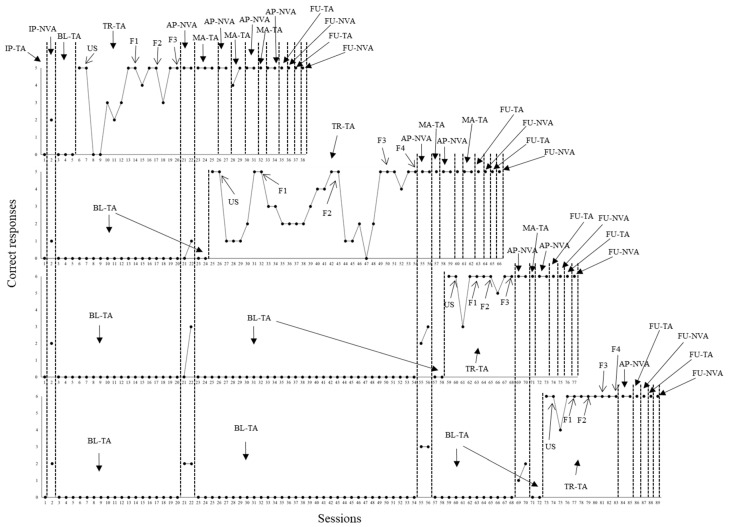
Correct responses in tact and non-verbal actions by P2. Note: Each of the four graphs above represent the performance of participant P2 concerning the emission of sentences with tact function and related non-verbal actions under the control of pictures from each of the behavioral sequences programmed (from first to last graph in Figure 1). Sequence data were organized in the following order from first to last graph: (1) cleaning up the hands, (2) putting on the mask and keeping it on the face, (3) functional play, and (4) brushing teeth. Different conditions were arranged throughout the sessions of each behavioral sequence. IP-TA—initial probe for tacts. IP-NVA—initial probe for non-verbal actions. BL-TA—baseline for tacts. TR-TA—tact training. First, P2 had to respond under the control of unfaded scripts (USs) until a learning criterion was achieved. Then, script fading was established in four or three fading steps (from F1 to F4 or F1 to F3, depending on the behavioral sequence). AP-NVA—additional probe for non-verbal actions. MA-TA—maintenance of tact training. FU-TA and FU-NVA mean follow up for tacts and non-verbal actions, respectively.

One initial tact probe session (IP-TA) and one initial related non-verbal action probe session (IP-NVA) were administered. P2, for all behavioral sequences, did not show correct responses in IP-TA. Regarding IP-NVA, a low performance between one and two correct responses was shown. During baseline for tact (BL-TA), no correct responses occurred across all behavioral sequences. During tact training with the first sequence (cleaning up the hands), two, seven, three, and three were needed to reach criterion in US, F1, F2, and F3, respectively. During AP-NVA probe sessions, no errors were committed regarding the first behavioral sequence. In the cases of the remaining sequences, the performance was low, ranging from one and three correct responses. In addition, still considering first sequence, independent performance for both repertoires was shown in additional probe sessions (maintenance tact training–probing non-verbal actions), as well as across follow up probes. At the same time, tact teaching was defined for the second behavioral sequence (putting on the mask and keeping it on the face).

During the teaching with the second sequence, two, six, eleven, seven, and four sessions were needed to reach criterion in US, F1, F2, F3, and F4. When AP-NVA probe sessions were conducted, no errors were committed with the second sequence. Regarding the third and fourth sequences, performance was low (ranging from two to three correct responses). Still considering the second sequence, no errors occurred for the repertoires across follow up probe sessions. In parallel, tact training was established with the third behavioral sequence (functional play).

During training with the third sequence, two, three, two and three sessions were needed to achieve the learning criterion in US, F1, F2, and F3. When AP-NVA probe sessions were applied, no errors occurred with the third sequence and performance was low with the fourth sequence. Still with the third sequence, independent performance for the repertoires was demonstrated in additional sessions (maintenance tact training–probing non-verbal actions), as well as across follow up probe sessions. In parallel, tact training was established with the fourth behavioral sequence (brushing teeth).

P2 needed two, three, two, two, and two sessions to achieve criterion in US, F1, F2, F3, and F4. When AP-NVA probe sessions were applied, P2 did not show errors with the fourth sequence. Independent performance with the repertoires was replicated across additional sessions (maintenance tact training–probing non-verbal actions), as well as no errors occurred across follow up probe sessions. Figure 3, next, shows data from P3.

**Figure 3 behavsci-15-00198-f003:**
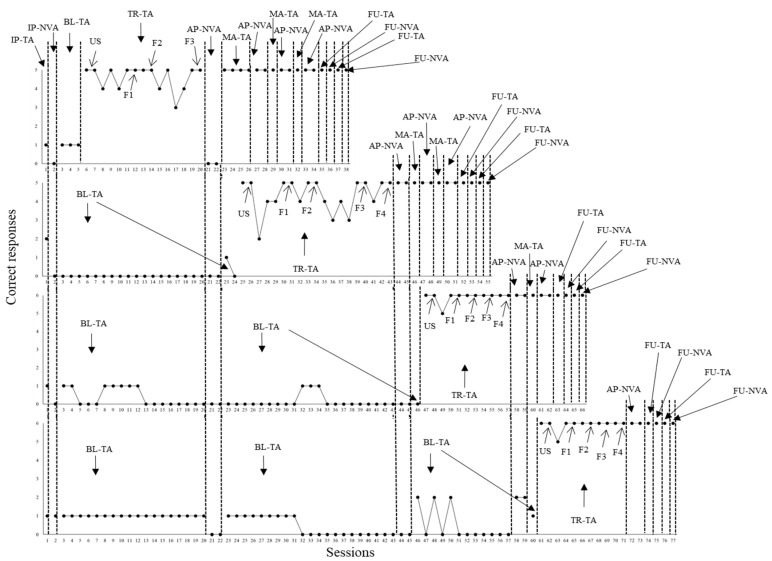
Correct responses in tact and non-verbal actions by P3. Note. Each of the four graphs above represent the performance of participant P3 concerning the emission of sentences with tact function and related non-verbal actions under the control of pictures from each of the behavioral sequences programmed (from first to last graph in Figure 1). Sequence data were organized in the following order from first to last graph: (1) cleaning up the hands, (2) putting on the mask and keeping it on the face, (3) functional play, and (4) brushing teeth. Different conditions were arranged throughout the sessions of each behavioral sequence. IP-TA—initial probe for tacts. IP-NVA—initial probe for non-verbal actions. BL-TA—baseline for tacts. TR-TA—tact training. First, P3 had to respond under the control of unfaded scripts (USs) until a learning criterion was achieved. Then, script fading was established in four or three fading steps (from F1 to F4 or F1 to F3, depending on the behavioral sequence). AP-NVA—additional probe for non-verbal actions. MA-TA—maintenance of tact training. FU-TA and FU-NVA mean follow up for tacts and non-verbal actions, respectively.

When one initial probe session for tact (IP-TA) and one initial probe session for related non-verbal actions (IP-NVAs) were administered, P3 demonstrated between only one and two correct tact responses concerning all four behavioral sequences. Likewise, only one correct non-verbal response occurred with the fourth sequence. During baseline for tact (BL-TA), overall, performance was low for all sequences. Across many sessions, P3 did not show beyond one correct response. Close to the end of the baseline with the fourth behavioral sequence, however, three correct responses were emitted in three sessions. When tact training was defined for the first behavioral sequence (cleaning up the hands), two, five, two, and six sessions were needed to achieve the learning criterion in US, F1, F2, and F3, respectively.

When AP-NVA sessions were administered, no correct responses were demonstrated across all four behavioral sequences. Nevertheless, after four maintenance tact training sessions, the related non-verbal repertoires emerged in two new AP-NVA probe sessions (considering the first behavioral sequence). Thereafter, independent performance in the repertoires were demonstrated in additional probe sessions (maintenance tact training–probing non-verbal actions), as well as in follow up probe sessions. Parallel to the acquisition of repertoires concerning the first behavioral sequence, tact training with the second sequence (putting on the mask and keeping it on the face) commenced for P3. Two, five, three, six, and three sessions were needed to achieve the learning criterion in US, F1, F2, F3, and F4, respectively. When AP-NVA probe sessions were applied, as can be noticed in Figure 3, no errors were committed concerning the second behavioral sequence, but no correct responses were shown with third and fourth sequences.

Still regarding the second sequence, independent performance for both repertoires were replicated in additional sessions (maintenance tact training–probing non-verbal actions), as well as it was the case of follow up probe sessions. Parallel to the acquisition of the repertoires for the second sequence, tact training with the third sequence (functional play) commenced. Two, three, two, two, and two sessions were needed to achieve criterion in US, F1, F2, F3, and F4, respectively. When two AP-NVA sessions were conducted, P3 showed no errors within the third sequence. In the case of the fourth sequence, performance was low with two correct responses. Still regarding the third sequence, independent performance in the repertoires was demonstrated across additional sessions (maintenance tact training–probing non-verbal actions), as well as it was shown across follow up probe sessions.

Parallel to the acquisition of repertoires for the third sequence, tact training with the fourth behavioral sequence (brushing teeth) commenced. P3 needed two, three, two, two, and two sessions to reach criterion in US, F1, F2, F3, and F4. When two AP-NVA probe sessions were conducted, no errors occurred. Independent performance concerning the target repertoires was also replicated across follow up probe sessions.

For all participants, the data collection process was carried out across several meetings. Each meeting lasted, on average, 40 min. Data collection with P1 occurred in 11 meetings. The total time spent with this participant during the research stages was 440 min on average. Data collection with P2 occurred in nine meetings. The total time with this participant during the research stages was 360 min. Data collection with P3 occurred in ten meetings. The total time spent with this participant during the research stages was 400 min.

## 4. Discussion

The previous literature on procedures comprising scripts and their fading (combined with differential reinforcement of independent performance) to teach vocal verbal communication using sentences, as well as the emission of important non-verbal behavioral chains (e.g., daily functional skills such getting dressed, brushing teeth, and playing functionally) in children diagnosed with ASD and other cases of learning disabilities, demonstrated that such procedures are effective in establishing the mentioned repertoires, contributing to greater autonomy and functionality in life ([8]; [9]; [13]; [17]; [18]; [19]).

The current research systematically replicated the study by [15] ([15]), so that script fading was used to establish the vocal verbal emission of sentences with tact function under the control of pictures from behavioral sequences. The procedure was effective for all three children with ASD who participated. In this sense, the results were like those from the previous study, which was considered a pioneer in this topic. In addition, the current study extended the investigation by assessing the effects of tact training on the emission of related non-verbal actions depicted by the pictures from the behavioral sequences. In probe sessions that preceded tact training for all participants, performance concerning the emission of non-verbal actions involved few or no correct responses. As the teaching of tact relations was defined for each participant, the emergence of related non-verbal actions was demonstrated across all four behavioral sequences. Two of the sequences (cleaning up the hands and putting on the mask and keeping it on the face) are especially important as preventive measures against infectious diseases such as COVID-19. Follow up probes, two weeks after the intervention was discontinued, showed that both non-verbal and verbal repertoires were still correctly performed by all learners. However, it is important that future studies examine the sustainability of these repertoires in the long term.

The data suggested that, for certain learners diagnosed with ASD, the acquisition of behavioral chains, regarding functional play and other functional skills (e.g., cleaning up the hands, using mask to prevent diseases, using the bathroom, etc.), may not be established solely through direct teaching with prompting hierarchy and differential reinforcement of independent performance as happened in previous studies ([8]; [17]; [18]; [19]). In other words, teaching the emission of sentences with tact function under the control of pictures from behavioral sequences may result in the emergence of related non-verbal repertoires without direct teaching (data from this research suggested that this happened for all participants). This seems to represent what the literature defines as correspondence between verbal and non-verbal behavior, or “say-do” correspondence ([16]), since the participants in the research successfully demonstrated the non-verbal actions from the four sequences in probes, as they were taught to make verbal descriptions about the pictures in the sequences.

On the other hand, a limitation of this research was that the opposite path, that is, teaching the repertoire of non-verbal actions under the control of pictures from the sequences and measuring the effects on the emergence of emitting related sentences with tact function, was not investigated. Whether correspondence between non-verbal and verbal behavior, or “do-say” correspondence, in children with ASD like those from the current study is possible or not remains an empirical question. It is important, in this sense, that future studies investigate this possibility ([16]). The literature on the topic of “do-say” correspondence has discussed that the procedure of differential reinforcement has successfully reduced the emission of inaccurate reports about past behaviors related to item interaction, toy play, and academic tasks ([4]; [5]; [7]). In other words, the procedure may reduce the possibility of lies. In a more recent investigation ([12]), researchers were concerned in developing strategies that could both strengthen the emission of honest reports by young children and reduce transgressions.

In the research ([12]), the children were first taught to emit honest reports about touching or not a disallowed item. Thereafter, the transgression of touching the not permitted item was decreased by making the children pause, put the item away or move away from it, and play with other toys in the environment. Three children (one typical and two diagnosed with ASD), aged between 6 and 7 years old, demonstrated an increase in the emission of honest reports and a decrease in transgressions to satisfying levels according to their caregivers. A future study, in which the effects of teaching behavioral sequences (through prompting hierarchy and differential reinforcement) are measured on the establishment of corresponding reports through sentences and without direct teaching, may represent an alternative way of teaching and strengthening the emission of honest reports and corresponding socially approved behaviors in children with ASD by caregivers.

Another limitation corresponded to the selection of children with ASD eligible to participate in the current study. Overall, seven children were assessed to demonstrate the repertoires of emitting sentences with tact function and related non-verbal actions under the control of pictures from the four sequences defined. Four children were dismissed because they were able to demonstrate all target repertoires without errors in initial probe sessions. However, the remaining three children defined as participants demonstrated few or no correct responses concerning both target repertoires in initial probe session and across baseline sessions. It is important that the investigation be extended across more children with ASD in future research, as a way of testing the generality of the procedures and results. In addition, not identifying participants, who could not show the emergence of non-verbal actions through the direct teaching of related tact relations, may also be interpreted as a limitation. Therefore, future investigations could involve participants less likely to show emergent related non-verbal repertoires, making it possible to assess if direct teaching procedures such as prompting hierarchy and differential reinforcement of independent performance will be effective, as it was the case in previous literature ([8]; [13]; [17]; [18]; [19]).

A limitation similar to what happened previously in [15] ([15]) was that a single sentence/script was defined for each of the pictures across the behavioral sentences. Although script fading was successful in establishing the emission of sentences free from script control in both studies (the previous and the current one as well), all participants showed the same sentences based on the scripts previously used during teaching. However, the literature discusses the importance of designing procedures that may favor the emission of varied sentences during communication, and that may not be only based on trained scripts ([9]; [11]). Therefore, new investigations may consider using varied sentences during script fading training, and assessing the impact on communication with varied sentences, including possible verbal utterances not directly based on previously trained scripts. A possible future systematic replication of the current research might address this issue. It will be important to assess how script fading training to establish tacts with varied sentences impact the emergence of related non-verbal repertoires.

This study did not assess generalization. One of the main results was the emergence of non-verbal repertoires through the training of verbal repertoires (tacts). Procedures, which tend to produce the acquisition of skills beyond what is directly taught to learners with ASD, are highly valuable. Regarding the possibility of assessing stimulus generalization, future studies could, for example, add probes to assess the emission of tacts under the control of different pictures representing the same behavioral sequences involved in the intervention. In addition, it is important that new studies also assess generalization across new environments and communication partners. This investigation also did not compare the research results with control groups, but it is acknowledged that this would be important. A future study could, for example, define a fifth behavioral sequence as control, and tact and related non-verbal repertoires regarding this new sequence could only be probed throughout the entirely new investigation (no teaching would be conducted concerning the fifth behavioral sequence). Once performance on both skills proved to be consistently low and did not change throughout the investigation, compared to other sequences involving the teaching of tacts, this would be important for the internal validity of the study procedures and results. This type of measurement may be important in helping to demonstrate the experimental control of IV over DV, expected from single case designs such as the case of the concurrent multiple baseline used in the current research.

This research did not predict participation of children with ASD with more significant deficits in verbal skills, considering, for example, tact relations. Future research may reflect on possible repertoires that can be taught as an alternative to teaching tact relations. Another limitation was that no social validity measures concerning parental support were obtained. It is important that future studies take data regarding this. Finally, it is important to argue that interventions, such as those from the current study on children diagnosed with ASD, may be also relevant to improve rehabilitation outcomes in other individuals such as, for example, patients with an acquired brain injury (e.g., mild stroke). According to [22] ([22]), cognitive function predicts rehabilitation outcomes after brain injury. They also said that behavioral interventions may improve rehabilitation outcomes. Implicit learning, associated with motor recovery, may be improved in patients who suffered from a mild stroke and have the implicit learning ability intact. It is possible that the previously mentioned ABA principles and procedures, such as reinforcement, use of scripts, and prompts and their gradual fading, help improve repertoires in individuals affected by mild acquired brain injury, so they may become less dependent on others to perform, for example, daily living skills such as dressing and toileting.

## 5. Conclusions

This research aimed to (1) assess the efficacy of textual or dictated scripts and their fading on the emission of sentences with tact function under the control of pictures from four behavioral sequences in three children diagnosed with ASD, and (2) once tact training with script fading was effective, its effects on the possible emergence of related non-verbal repertoires for all sequences would be probed. Script fading was successful in establishing sentences with tact function for all participating children, as well as related non-verbal responses were demonstrated, suggesting their emergence due to tact training. This outcome was interpreted as a possible correspondence between verbal and non-verbal repertoires or, in other words, “say-do” correspondence. Limitations of the research were pointed out, as well as recommendations of future research to address them.

Data from this research suggest that the establishment of important behavioral chains/sequences that make children with ASD more independent and functional do not need to be directly taught. Instead, depending on the learner, these non-verbal repertoires may be derived or emergent from the teaching of related verbal repertoires, as appears to have been the case in the present study. The script fading procedure appears to have contributed to this result.

## Figures and Tables

**Figure 1 behavsci-15-00198-f001:**
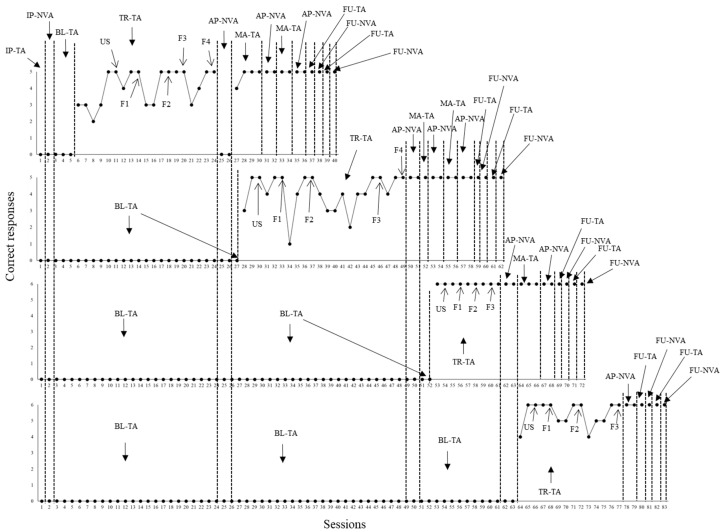
Correct responses in tact and non-verbal actions by P1. Note: Each of the four graphs above represent the performance of participant P1 concerning the emission of sentences with tact function and related non-verbal actions under the control of pictures from each of the behavioral sequences programmed (from first to last graph in Figure 1). Sequence data were organized in the following order from first to last graph: (1) cleaning up the hands, (2) putting on the mask and keeping it on the face, (3) functional play, and (4) brushing teeth. Different conditions were arranged throughout the sessions of each behavioral sequence. IP-TA—initial probe for tacts. IP-NVA—initial probe for non-verbal actions. BL-TA—baseline for tacts. TR-TA—tact training. First, P1 had to respond under the control of unfaded scripts (USs) until a learning criterion was achieved. Then, script fading was established in four or three fading steps (from F1 to F4 or F1 to F3, depending on the behavioral sequence). AP-NVA—additional probe for non-verbal actions. MA-TA—maintenance of tact training. FU-TA and FU-NVA mean follow up for tacts and non-verbal actions, respectively.

**Table 1 behavsci-15-00198-t001:** Items and scripts for each behavioral sequence.

Behavioral Sequence	Items	Scripts
Sequence 1—Cleaning up the hands	Bottle containing hand sanitizer Pictures representing the step-by-step process for carrying out the task	Action 1: “Take the bottle with hand sanitizer” Action 2: “Apply the hand sanitizer to your hand” Action 3: “Rub one hand Against the other hand” Action 4: “Wait for your hands to dry” Action 5: “Once dry, your hands are clean”
Sequence 2—Putting on the mask and keeping it on the face	Disposable mask for private use of the participating child Pictures representing the step-by-step process for carrying out the task	Action 1: “Pick up the mask by holding it by the straps” Action 2: “Place the mask Against your face, covering your nose and mouth” Action 3: “Place the straps behind your ears” Action 4: “Adjust the mask to your face” Action 5: “Keep the mask on”
Sequence 3—Functional play	Mr. Potato Head doll Pictures representing the step-by-step process for carrying out the task	Action 1: “Fit the hat to Mr. Potato Head” Action 2: “Attach the eyes to the Potato Head” Action 3: “Fit the nose to Mr. Potato Head” Action 4: “Fit the mouth to Mr. Potato Head” Action 5: “Fit the ears to Mr. Potato Head” Action 6: “Fit the Arms to Mr. Potato Head”
Sequence 4—Brushing teeth	Toothbrush and toothpaste for the participant’s personal use Pictures representing the step-by-step process for carrying out the task	Action 1: “Take the toothbrush, turn on the tap and wet the toothbrush” Action 2: “Put toothpaste on the toothbrush” Action 3: “Brush your front, bottom and top teeth” Action 4: “Brush your tongue” Action 5: “Wash your mouth and your toothbrush” Action 6: “Store the toothbrush”

**Table 2 behavsci-15-00198-t002:** Research stages.

Stages	Condition with Textual Scripts (P2)	Condition with Dictated Scripts for the Emission of Echoics (P1 and P3)
First stage	Initial probe and baseline of emission of sentences with tact function and initial probe for the emission of non-verbal related responses	Initial probe and baseline of emission of sentences with tact function and initial probe for the emission of non-verbal related responses
Second stage	Teaching script reading	Teaching the emission of echoics under the control of dictated scripts
Third stage	Script fading (fading of textual scripts)	Script fading (fading of dictated scripts)
Fourth stage	Additional probes to assess the emergence of the emission of non-verbal responses concerning the behavioral sequences	Additional probes to assess the emergence of the emission of non-verbal responses concerning the behavioral sequences
Fifth stage	Follow up (two weeks after the end of research)	Follow up (two weeks after the end of research)

## Data Availability

The authors make the raw data underlying the conclusions in this article available upon request.

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
