# Peer review of "Teaching Communication and Functional Life Skills in Children Diagnosed with Autism Spectrum Disorder"

_behavsci, 2025, doi:10.3390/bs15020198_

Round 1

Reviewer 1 Report

Comments and Suggestions for Authors

Dear authors, thank you for the opportunity to review your work. It is a relevant work that provides evidence for teaching communication skills to children with Autism Spectrum Disorder who have limitations in the use of language in everyday situations. The results obtained have been of great interest to me and I intend to use them.

The proposed modifications must be considered minor and are at the discretion of the authors. Basically, remove the information about the intervention procedure from the end of the introduction section and include it in the procedure section (Experimental Design), as well as indicate the duration of the procedure (by stages) for each participant.

Author Response

Cover letter

Dear reviewer,

Thank you very much for the important considerations. As may be seen below, we have carried out changes to the manuscript according to the demands.

Reviewer 1

Comments and Suggestions for Authors

Dear authors, thank you for the opportunity to review your work. It is a relevant work that provides evidence for teaching communication skills to children with Autism Spectrum Disorder who have limitations in the use of language in everyday situations. The results obtained have been of great interest to me and I intend to use them.

The proposed modifications must be considered minor and are at the discretion of the authors. Basically, remove the information about the intervention procedure from the end of the introduction section and include it in the procedure section (Experimental Design), as well as indicate the duration of the procedure (by stages) for each participant.

  • We have removed the detailed information regarding intervention from the end of the introduction and added it to the section on the experimental design used.
  • In the end of the result section, we explained that, for all participants, the data collection process was carried out across several meetings. Each meeting lasted, on average, 40 minutes. Data collection with P1 occurred in 11 meetings. The total time with this participant during data collection was 440 minutes on average. Data collection with P2 occurred in nine meetings. The total time with this participant during data collection was 360 minutes on average. Data collection with P3 occurred in ten meetings. The total time with this participant during data collection was 400 minutes on average.

Reviewer 2 Report

Comments and Suggestions for Authors

We thank the editor and authors, Juliana Ribeiro Rabelo Costa and Daniel Carvalho de Matos, for the opportunity to review the manuscript titled "Teaching Communication and Functional Life Skills in Children Diagnosed with Autism Spectrum Disorder." The topic addressed is highly relevant in the field of applied behavior analysis, significantly contributing to the existing literature on strategies to enhance communication and functional skills in children with ASD.

Strengths of the article:

  • Rigorous methodology with well-defined experimental designs (lines 209–211).
  • Precise implementation of script fading, shown to be effective in the results (lines 556–560).
  • Clear explanation of procedures, ensuring reproducibility of the study (lines 221–225).
  • Attention to detail in the materials used (lines 333–342).

Modifications to be made (line by line):

  1. Line 88: "Scripts may be visual, textual or auditory." → Consider providing examples for greater clarity.
  2. Line 189: "Generalization measures were obtained..." → Elaborate on the criteria for generalization.
  3. Line 333: "Non-verbal stimuli were used..." → Include representative images as an online supplement.
  4. Line 544: "Script fading produced the emission..." → Explain why these results differ from previous studies.
  5. Line 750: Add a reflection on the utility of fading for other populations.

Sections to remove or simplify:

  • Lines 98–113: The detailed discussion of previous studies could be condensed to avoid redundancy.
  • Lines 1216–1230: The information is interesting but seems to reiterate previously mentioned concepts.

Evaluation of data and results:
The data is clearly presented, with well-organized graphs (lines 650–700). However, it is suggested to delve deeper into the results related to generalization (lines 560–600). Additionally, comparing the results with control groups would be beneficial.

Unmentioned limitations identified:

  1. The sustainability of behaviors in the long term was not examined.
  2. Participants with lower skill levels were not tested, limiting generalizability.
  3. Details on parental support to continue using the strategies are missing.

Citation suggestion:
Include the following article as a theoretical reference on the validation of communication tools:

  • Diotaiuti, P., Valente, G., Mancone, S., & Grambone, A. (2020). Psychometric Properties and a Preliminary Validation Study of the Italian Brief Version of the Communication Styles Inventory (CSI-B/I). Frontiers in Psychology, 11, 1421. DOI: https://doi.org/10.3389/fpsyg.2020.01421.
    Suggest citing this article in the Introduction, specifically after "establish socially significant repertoires" (line 127), to strengthen the theoretical context.

Author Response

Cover letter

Dear reviewer,

Thank you very much for the important considerations. As may be seen below, we have carried out changes to the manuscript according to the demands.

Reviewer 2

Comments and Suggestions for Authors

We thank the editor and authors, Juliana Ribeiro Rabelo Costa and Daniel Carvalho de Matos, for the opportunity to review the manuscript titled "Teaching Communication and Functional Life Skills in Children Diagnosed with Autism Spectrum Disorder." The topic addressed is highly relevant in the field of applied behavior analysis, significantly contributing to the existing literature on strategies to enhance communication and functional skills in children with ASD.

Strengths of the article:

  • Rigorous methodology with well-defined experimental designs (lines 209–211).
  • Precise implementation of script fading, shown to be effective in the results (lines 556–560).
  • Clear explanation of procedures, ensuring reproducibility of the study (lines 221–225).
  • Attention to detail in the materials used (lines 333–342).

Modifications to be made (line by line):

  1. Line 88: "Scripts may be visual, textual or auditory." → Consider providing examples for greater clarity.
  • We have provided examples in parentheses
  1. Line 189: "Generalization measures were obtained..." → Elaborate on the criteria for generalization.
  • We have explained that, for one child, training took place in a clinic room. For the remaining two children, training occurred in the home setting. After intervention was successful for each learner in an environment, generalization measures, across the defined tasks, were obtained through probes in a new environment (either clinic or home depending on the child).
  1. Line 333: "Non-verbal stimuli were used..." → Include representative images as an online supplement.
  • We have uploaded a file containing all pictures used to teach the vocal verbal emission of sentences with tact function and to probe the emission of related non-verbal actions.
  1. Line 544: "Script fading produced the emission..." → Explain why these results differ from previous studies.
  • We have explained that our study replicated a previous one in which the same procedure (script fading) was used with the purpose of establishing the emission of sentences with tact function (according to a Skinnerian approach of language). Other investigations into script fading were not specifically concerned with teaching tacts and probing the emergence of related untaught repertoires. Besides, we argued that our study extended the previous one (aimed to establish tacts) by assessing the emergence of related non-verbal actions.
  1. Line 750: Add a reflection on the utility of fading for other populations.
  • We argued that interventions, such as those from the current study on children diagnosed with ASD, may be also important to improve rehabilitation outcomes in patients with an acquired brain injury (e.g., mild stroke). According to Whyte et al. (2011), cognitive function predicts rehabilitation outcomes after brain injury. They also said that behavioral interventions may improve rehabilitation outcomes. Implicit learning, associated with motor recovery, may be improved in patients who suffered from a mild stroke and have the implicit learning ability intact. It is possible that the previously mentioned ABA principles and procedures, such as reinforcement, use of scripts, prompts and their gradual fading, help improve repertoires in individuals affected by mild acquired brain injury, so they may become less dependent on others to perform, for example, daily living skills such as dressing and toileting.

Sections to remove or simplify:

  • Lines 98–113: The detailed discussion of previous studies could be condensed to avoid redundancy.
  • Dear reviewer, we sought to condense information related to previous studies, but not those concerning the use of script fading to teach communication using sentences, especially the case aimed to teach communication with tact function (according to a Skinnerian taxonomy of behavioral language analysis) (Matos et al., 2022). Our investigation consisted of a systematic replication of this study. Besides, if teaching children with ASD to emit sentences with tact function, to describe pictures from behavioral sequences, did not result in the emergence of related untaught non-verbal repertoires, we would teach these repertoires in a second experiment seeking to systematically replicate previous studies which involved the use of a least-to-most prompting hierarchy (Phillips & Vollmer, 2012; Phillips et al., 2019). This, however, was not necessary. Anyway, in the introduction section, we considered it important to, at least, describe the prompts used by Phillips and Vollmer (we removed those from the following study since they were similar). We also removed, or condensed, information that we considered less relevant to our investigation.
  • Lines 1216–1230: The information is interesting but seems to reiterate previously mentioned concepts.
  • Dear reviewer, we had difficulty trying to identify the part of the text related to the argument presented above. We configured line numbering in the text editor, but there did not seem to be a match. Anyway, we read the discussion section very carefully again and if possible, please, we would like to keep it unaltered. Besides, we added more information in this section regarding other important topics you have presented in your review.

Evaluation of data and results:
The data is clearly presented, with well-organized graphs (lines 650–700). However, it is suggested to delve deeper into the results related to generalization (lines 560–600). Additionally, comparing the results with control groups would be beneficial.

  • We did not assess generalization in our study. We probed two types of repertoires before intervention:1) the emission of sentences with tact function under the control of pictures from four behavioral sequences; 2) the emission of related non-verbal repertoires regarding the sequences. During intervention, the tacts (first type of repertoire mentioned) concerning each sequence were systematically taught through script fading to each participant. Throughout the teaching of tacts, additional probes to verify the possible emergence of related non-verbal repertoires were administered. One of the main results of this study was the emergence of these non-verbal repertoires, which never needed to be taught to all learners. Procedures, which tend to produce acquisition of skills beyond what is directly taught to learners with ASD, are highly valuable. Regarding the possibility of assessing stimulus generalization, future studies could, for example, add probes to assess the emission of tacts under the control of different pictures representing the same behavioral sequences involved in the intervention. We also did not compare the research results with control groups, but we acknowledge that this would be important. A future study could, for example, define a fifth behavioral sequence as control, and tact and related non-verbal repertoires regarding this new sequence could only be probed throughout the entire new investigation (no teaching would be conducted concerning the fifth behavioral sequence). Once performance on both skills proved consistently low and did not change throughout the investigation, compared to other sequences involving the teaching of tacts, this would be important for the internal validity of the study procedures and results. This type of measurement can be important in helping to demonstrate the experimental control of IV over DV, expected from single case designs such as the case of the concurrent multiple baseline used in the current research.

Unmentioned limitations identified:

  1. The sustainability of behaviors in the long term was not examined.
  • We discussed that follow up probes, two weeks after the intervention was discontinued, showed that both non-verbal and verbal repertoires were still correctly performed by all learners. However, it is important that future studies examine the sustainability of these repertoires in the long term.
  1. Participants with lower skill levels were not tested, limiting generalizability.
  • We agree. This was emphasized in the discussion section.
  1. Details on parental support to continue using the strategies are missing.
  • In fact, we did not obtain social validity measures concerning parental support. This was a limitation of the research.

Citation suggestion:
Include the following article as a theoretical reference on the validation of communication tools:

  • Diotaiuti, P., Valente, G., Mancone, S., & Grambone, A. (2020). Psychometric Properties and a Preliminary Validation Study of the Italian Brief Version of the Communication Styles Inventory (CSI-B/I). Frontiers in Psychology, 11, 1421. DOI: https://doi.org/10.3389/fpsyg.2020.01421.
    Suggest citing this article in the Introduction, specifically after "establish socially significant repertoires" (line 127), to strengthen the theoretical context.

  • Dear reviewer, thank you for the suggestion. We have included the mentioned article.  

Round 2

Reviewer 2 Report

Comments and Suggestions for Authors

After careful review and consideration of the manuscript titled "The Influence of Personality Traits on College Students' Exercise Behavior: A Chain Mediation Model of Exercise Self-Efficacy and Exercise Motivation," I am pleased to inform you that your work is now ready to move to the next stage towards publication.

The manuscript has demonstrated thorough theoretical understanding and solid empirical analysis, making a significant contribution to the existing literature. The revisions made in response to the previous review comments have significantly improved the clarity and robustness of the study.

I encourage you to proceed with the final editorial steps necessary for publication. Please ensure to adhere to the formatting guidelines and verify all citations and references to maintain the academic integrity of the work.

Congratulations on the hard work and commitment shown in continually improving your manuscript. We look forward to the publication of your study, which will undoubtedly stimulate further research and discussion in the field.